# One mean to rule them all? The arithmetic mean based egg reduction rate can be misleading when estimating anthelminthic drug efficacy in clinical trials

**Wendelin Moser[1,2], Jennifer Keiser[1,2], Benjamin Speich[3,4], Somphou Sayasone[2,5,6], Stefanie Knopp[2,5], Jan Hattendorf[2,5] \***

**1** Department of Medical Parasitology and Infection Biology, Swiss Tropical and Public Health Institute, Basel, Switzerland, **2** University of Basel, Basel, Switzerland, **3** Centre for Statistics in Medicine, Nuffield Department of Orthopaedics, Rheumatology and Musculoskeletal Sciences, University of Oxford, Oxford, United Kingdom, **4** Basel Institute for Clinical Epidemiology and Biostatistics, Department of Clinical Research, University Hospital Basel, University of Basel, Basel, Switzerland, **5** Department of Epidemiology and Public Health, Swiss Tropical and Public Health Institute, Basel, Switzerland, **6** Lao Tropical and Public Health Institute, Vientiane, Lao People's Democratic Republic

\* jan.hattendorf@swisstph.ch

**Data Availability Statement:** The data cannot be shared without restrictions because the authors do not own the data. The data underlying the results

## Abstract

Animal and human helminth infections are highly prevalent around the world, with only few anthelminthic drugs available. The anthelminthic drug performance is expressed by the cure rate and the egg reduction rate. However, which kind of mean should be used to calculate the egg reduction rate remains a controversial issue. We visualized the distributions of egg counts of different helminth species in 7 randomized controlled trials and asked a panel of experts about their opinion on the egg burden and drug efficacy of two different treatments. Simultaneously, we calculated infection intensities and egg reduction rates using different types of means: arithmetic, geometric, trimmed, winsorized and Hölder means. Finally, we calculated the agreement between expert opinion and the different means. We generated 23 different trial arm pairs, which were judged by 49 experts. Among all investigated means, the arithmetic mean showed poorest performance with only 64% agreement with expert opinion (bootstrap confidence interval [CI]: 60−68). Highest agreement of 94% (CI: 86−96) was reached by the Hölder mean $M_{0.2}$, followed by the geometric mean (91%, CI: 85−94). Winsorized and trimmed means showed a rather poor performance (e.g. winsorization with 0.1 cut-off showed 85% agreement, CI: 78−87), but they performed reasonably well after excluding treatment arms with a small number of patients. In clinical trials with moderate sample size, the currently recommended arithmetic mean does not necessarily rank anthelminthic efficacies in the same order as might be obtained from expert evaluation of the same data. Estimates based on the arithmetic mean should always be reported together with an estimate, which is more robust to outliers, e.g. the geometric mean.

presented in the study are available from the authors of the original studies [ref. 15, 21-25].

**Funding:** WM and JK were partly funded by Swiss National Science Foundation (No 320030_14930). The funders had no role in study design, data collection and analysis, decision to publish, or preparation of the manuscript.

**Competing interests:** The authors have declared that no competing interests exist.

## Author summary

Besides cure rates, egg reduction rates represent an important indicator of anthelminthic drug efficacy in clinical trials. However, there is an ongoing controversy whether the arithmetic or the geometric mean should be used for its calculation. The arithmetic mean is problematic in skewed distributions mainly because the mean is sensitive to outliers, whereas the geometric mean does not correspond to our intuitive interpretation of average reduction. Several studies tried to compare the performance of different means but they relied on assumptions, which favored one approach over another. Despite the ongoing debate, the World Health Organization (WHO) recommends the arithmetic mean to calculate egg reduction rates. To overcome limitations from previous studies, we visualized data from several clinical trials and asked a panel of experts to compare drug efficacy of two different treatments. Afterwards, we estimated efficacy by using different means. Finally, we calculated the raw agreement between expert opinion and the different means. From all investigated methods to calculate efficacy, the arithmetic mean showed the poorest performance in terms of agreement with expert opinion. In anthelminthic human drug trials, which are characterized by small sample size and non-adherence, estimates more robust to outliers should be reported to assess drug efficacy performance.

## Introduction

Helminths, including cestodes, nematodes and trematodes, infect a large number of humans and animals. Among humans, helminth infections are highly prevalent with for example, 1.5 billion people infected with soil-transmitted helminths (STHs, *Ascaris lumbricoides*, hookworm and *Trichuris trichiura*) [1], 240 million with schistosomes [2] and 120 million with lymphatic filaria [3]. In livestock production, helminth infections are responsible for decreased productivity, which leads to economic losses for famers [4]. To control human helminth infections, the World Health Organization's (WHO) goal is to reduce the burden caused by moderate and heavy infections by increasing the coverage of anthelminthic drugs within so-called preventive chemotherapy programs–i.e. annual or biannual mass treatment of high risk populations [5]. Anthelminthic resistance has been observed widely in veterinary medicine [6–8]; therefore, emergence of resistance in humans is likely [9,10]. Hence, it is crucial to closely observe the anthelminthic drug efficacy in order to detect resistance development [11,12].

From a clinical medicine point of view cure rates (CRs) are usually of primary interest; however, from a public health perspective and for monitoring drug resistance, egg reduction rates (ERRs) are often more appropriate compared to CRs [13] and are therefore commonly used in human and exclusively used in veterinary medicine [12,14]. The ERR is defined as the relative reduction in the group mean egg output after treatment compared to pre-treatment levels. For estimating the ERR two types of means–or, more precisely, measures of central tendency–are exclusively used: the arithmetic mean and the geometric mean. Both means have strengths and weaknesses, triggering an ongoing debate among researchers and disease control specialists, which measure to prefer. One main disadvantage of the arithmetic mean is the influence of outliers. An example was reported by Speich and colleagues [15]; one extreme outlier resulted in a decrease in ERR from 93% to 73%. In addition, the arithmetic mean is not in close proximity to most of the observations in skewed distributions. To reduce the influence of outliers for skewed parasite data, commonly the geometric mean is used. Its disadvantages include the assumption of homogeneity of the variance between the compared groups [16,17]

and the arbitrary choice of the constant for taking the logarithm of zero egg counts at follow-up [18]. The current WHO guidelines recommend the use of the arithmetic mean for calculating ERRs [13].

Several researchers have continued to identify the most appropriate method for calculating ERRs using empirical data or computer simulations. However, the methods used were based on assumptions about true efficacy or egg distribution, which favored one specific mean over another [17–20]. For instance, if we define the performance of a mean as the unbiased estimation of the relative egg reduction in the population, the arithmetic mean will outperform the geometric mean in any distribution without extreme values. Conversely, if we define performance as sensitivity to outliers, range of the confidence interval or proximity to the median, the geometric mean will always show a better performance.

This study applied a new approach to assess the performance of different means to calculate ERRs. To overcome previous shortcomings, we visualized the distributions of egg counts of different helminth species in selected randomized controlled trials and asked a panel of experts about their opinion on the egg burden and drug efficacy of two different treatments. Afterwards, we calculated means and ERRs based on different types of means and assessed their agreement with the expert opinion. Of note, we used for this study exclusively data from human drug trials with small to moderate sample size (range 13 to 140 participants per arm). The results should not be extrapolated to other scenarios.

## Methods

The methods can be divided into four main steps: i) gathering and preparing data from previously conducted randomized multi-arm anthelminthic drug trials and dividing the trial arms into pairs, ii) visualizing the egg count distributions and asking experts for their opinion, which one of the two trial arms has a higher egg burden (before and after treatment) and better drug efficacy, iii) calculating mean egg counts at baseline and follow-up and ERRs of each trial arm using different types of means, and iv) assessing the performance of each type of mean according to their proportional agreement with the experts. The steps are summarized in Fig 1.

### Data preparation

Seven clinical drug trials against helminths with a total of 33 study arms, for which individual patient level data was available in house, were used for generating the questionnaires for experts [15, 21–25]. If efficacy was reported for more than one helminth species, all species were included resulting in a total of 46 arms. The trial arms were, stratified by study and species, ordered according to arithmetic mean infection intensity and grouped into consecutive pairs.

### Expert opinion on egg counts and drug efficacy

**Questionnaire format.**   For each of the 23 trial arm pairs we generated several figures visualizing the egg count distributions with box-plots and kernel density plots (the latter can be interpreted as a histogram but is displayed as a continuous line instead of bars). The plots were separately generated for baseline and follow-up and were represented on linear and log scale using R's stats package with default settings (except for the smoothing bandwidth of the density plots, which was set as the maximum egg counts of both trial arms divided by 20). A constant of 1 was added to the egg counts before logarithmic transformation. The experts were asked to judge if the egg burden is considerably higher in arm A, slightly higher in A, similar, slightly higher in B, or considerably higher in B separately for baseline and follow up. Similarly,

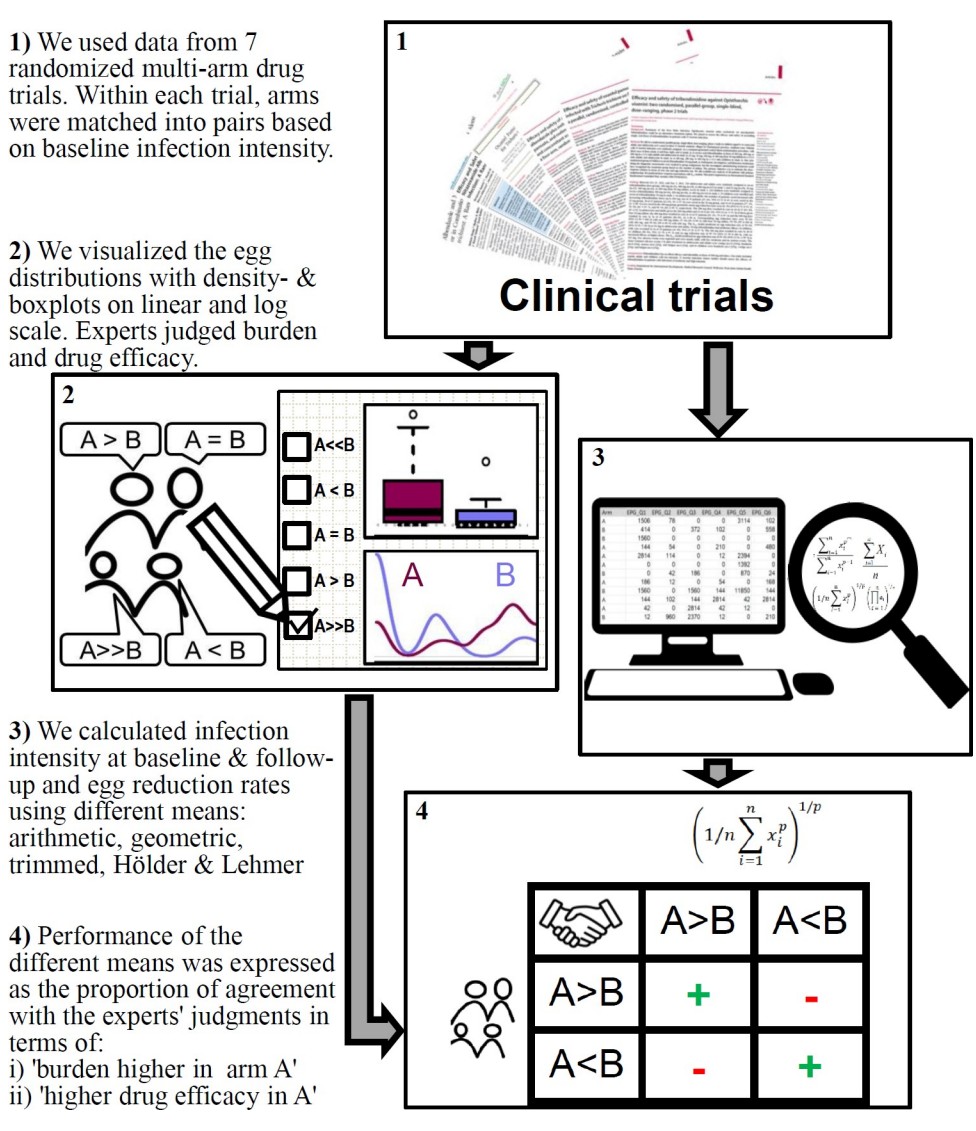

**1)** We used data from 7 randomized multi-arm drug trials. Within each trial, arms were matched into pairs based on baseline infection intensity.

**2)** We visualized the egg distributions with density- & boxplots on linear and log scale. Experts judged burden and drug efficacy.

**3)** We calculated infection intensity at baseline & follow-up and egg reduction rates using different means: arithmetic, geometric, trimmed, Hölder & Lehmer

**4)** Performance of the different means was expressed as the proportion of agreement with the experts' judgments in terms of:
i) 'burden higher in arm A'
ii) 'higher drug efficacy in A'

**Fig 1. Illustration of the study design.**

we asked for their opinion about treatment efficacy whereby the following options were provided: Treatment A is better, A slightly better, Similar, B slightly better, and B better. We generated several questionnaires and, in each questionnaire, the order of questions and the allocation of trial arms to A and B were randomly shuffled. One questionnaire example is presented in S1 File.

**Questionnaire distribution.** Experts including biostatisticians, human parasitologists and epidemiologists, and veterinary parasitologists and epidemiologists with long-term experience in helminthic diseases selected from personal contacts of the authors were asked to fill in this questionnaire between February and November 2016. All participants were asked for additional contacts of potential specialists to increase the number of participants. The questionnaires were distributed either via a hard copy or sent by email. After the distribution, each participant received up to five reminders. Participants were asked for their personal interpretation of the data and they were informed that there is no right or wrong answer.

## Calculation of mean egg counts and ERRs with different means

The geometric mean is defined as:

$$GM(x_1, \ldots, x_n) = e^{(1/n \sum_{i=1}^{n} \log(x_i))} \tag{1}$$

The geometric mean requires $x1, \ldots, x_n > 0$. Therefore, a small amount (usually 1) has to be added to account for zero egg counts. The amount is usually subtracted from the final results:

$$GM(x_1, \ldots, x_n) = e^{(1/n \sum_{i=1}^{n} \log(x_i+1))} - 1 \tag{2}$$

The Hölder mean (syn. power mean) is defined as:

$$H_p(x_1, \ldots, x_n) = (1/n \sum_{i=1}^{n} x_i^p)^{1/p} \tag{3}$$

with parameter $p \neq 0$ and $x_1, \ldots, x_n \geq 0$. The arithmetic and geometric means are special cases of the Hölder mean with $p = 1$ and $p = \lim_{p \to 0}$, respectively. Another common mean is the Lehmer mean defined as:

$$L_p(x_1, \ldots, x_n) = \frac{\sum_{i=1}^{n}(x_i + 1)^p}{\sum_{i=1}^{n}(x_i + 1)^{p-1}} - 1 \tag{4}$$

Just like the geometric mean, the Lehmer mean requires values $> 0$. Therefore, also in this case 1 is added to account for zero counts.

The truncated and winsorized means are less sensitive to extreme values. For the truncated mean a certain percentage of the ends are discarded whereas for the winsorized mean the values are replaced by the most extreme remaining values. Several algorithms exist to determine quantiles, we used the inverse of the empirical distribution function with averaging at discontinuities (type 2 in R, type 5 in SAS). This quantile algorithm–in contrast to several others–satisfies $M(e) = M(\{e,e\})$ for each n-tuple $e$ of n elements. Truncation and winsorization is normally applied at both ends, but–because we are only worried about extremely high egg counts–we discarded/replaced only the highest values.

In total we calculated mean egg counts and ERRs for 30 different means: arithmetic and geometric means, Hölder and Lehmer means with parameter p set to 0.1, 0.2, ..., 0.9 and winsorized and truncated means with discarding, replacing, 2, 4, 6, 8 and 10% of the highest values.

## Assessing the performance of each mean as agreement with experts

To assess agreement between experts and calculated means we dichotomized both variables. For the calculated means we simply used the difference between both arms to decide of arm A or arm B showed higher egg counts or egg reductions, ignoring the magnitude of the difference. Expert opinion was dichotomized into the same categories based on two different definitions. i) 'all studies' (simple majority criterion): if more experts judged the egg burden/drug efficacy higher in a certain arm (e.g. number of persons answering either A is better and A is slightly better) compared to the number of experts favoring the other arm while ignoring the undecided. We used the score (arithmetic mean of the answers of the Likert scale transformed into numerical scores) to break ties. If the score was 3, which occurred once in the baseline and once in the follow-up judgments, the questions were excluded from the analysis. ii) 'consensus studies' (absolute majority criterion): more experts ($>50\%$) shared the view that the egg burden/drug efficacy is higher in a certain arm than undecided or those with an opposite view

together. We refer to this as 'consensus studies' because no or only very few experts had an opposite opinion (median = 0, range 0 to 3).

Additionally, we inspected visually the relationship between the calculated differences among the trial arm pairs and the raters score. Further, we explored the relationship between the calculated differences in ERRs and rater scores among the trial arm pairs and the difference of the observed CRs. Further information how agreement and performance was assessed is shown in the guidance S2 File.

### Data analysis

All data were analysed with R version 3.4.3. Performance was calculated as the raw percentage agreement between experts and each mean. Of note, raw agreement is an appropriate measure in our study because, by design, chance agreement is always 50%. Confidence intervals for agreement were constructed by bootstrap resampling of raters. A sensitivity analysis was conducted, which only included trial arms with a minimum number of observations of 30 per arm.

Inter-rater reliability among expert opinion was assed by Krippendorff's α for ordinal metrics. We also estimated the intra cluster correlation (single unit, random raters) and the average pairwise kappa coefficient with weighted squared distances. Because all estimates were very close, we present only Krippendorff's α. Loess smoothing lines were estimated with a smoothing parameter α of 0.85.

## Results

### Characteristics of included studies

Data from six publications [15,21–25] including seven clinical trials with 32 trial arms were used for generating the questionnaires. Among the clinical trials, four included drug efficacy data for treating *T. trichiura* [15,21–24], three for hookworm [21,22,24] and two for *O. viverrini* [25]. Different drugs, doses or drug combinations were used in the trials, i.e. albendazole [15,22–24], mebendazole [15,22,24], oxantel pamoate [15,21,22], ivermectin [15,24], nitazoxanide [23] and tribendimidine [25]. The median number of participants per arm was 48 (interquartile range: 39–112, range: 13–140). The median CR was 34% (range: 0–91%). Further trial arm characteristics including egg counts and cure rates are presented in S1 Table in S3 File).

### Response rate and field specifications

From a total of 76 invited experts, we received 49 (64.5%) filled-out questionnaires. Participants included human parasitologists/epidemiologists (n = 26, 53.1%), followed by veterinary parasitologists/epidemiologists (n = 12, 24.5%), biostatisticians (n = 9, 18.4%) and two engineers with experience in human parasitology (n = 2, 4.1%). The distribution of academic qualifications was as follows: 27 (55.1%) had a PhD-degree, 16 (32.7%) were professors, four had a MSc-degree (8.2%) and two participants were medical doctors (4.1%).

### Inter rater reliability of experts' judgments

The responses obtained for each question are visualized in Fig 2. As expected, the answer "Egg burden is similar" was quite common at baseline whereas a clear preference was found for the follow up and efficacy ratings. Krippendorff's α was estimated at 0.44, 0.62 and 0.65 for baseline, follow-up and efficacy, respectively. In 3.5% (40/1127) of the answers, the raters stated that they are not able to provide a reliable judgment. From the 69 comparisons, 37 (54%) fulfilled the absolute majority criterion, i.e. more than 50% of experts favor one arm and 67 (97%) fulfilled the simple majority criterion.

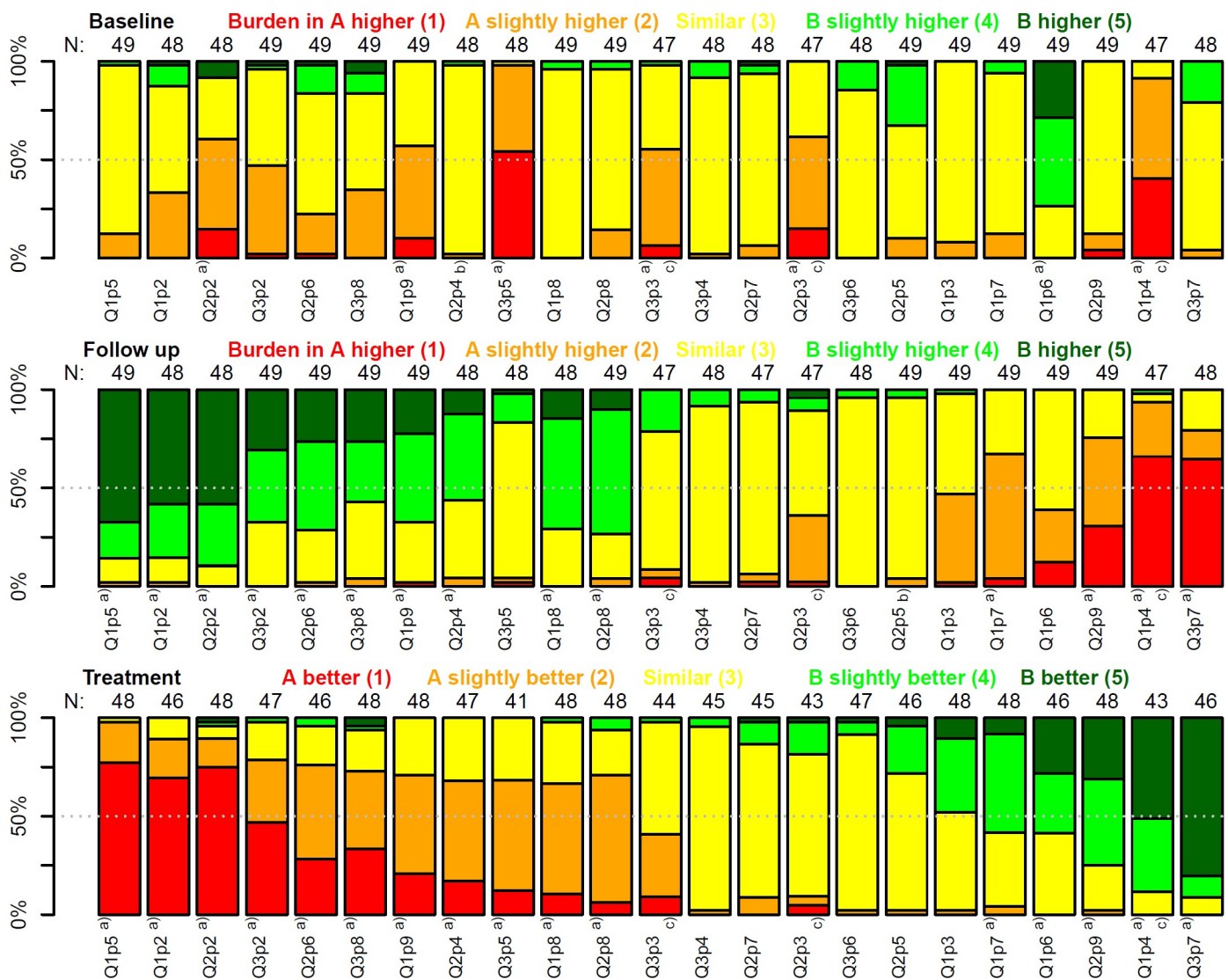

**Fig 2. Judgment of experts with respect to the egg burden and treatment efficacy of 2 clinical trial arms.** The labels below the bars denote the page and question (1: question at the top of the page, 2: middle, 3: bottom) in the example questionnaire presented in the S1 File. Numbers above bars represent the number of experts with a valid response (i.e. excluding "don't know" responses). Abbreviations: Q: Question; p: page. Top panel: baseline, middle panel: follow-up, bottom panel: efficacy. [a] consensus agreement (absolute majority criterion—more than 50% of experts favor one arm) [b] arm pair excluded, experts did not favor any arm [c] excluded from the sensitivity analysis (number of trial participants in 1 arm below 30).

## Performance of different means

The agreements between the different means and the expert opinion are presented in Fig 3. The arithmetic mean showed the poorest performance among all means. Especially, for comparisons at follow-up the agreement was close to chance agreement. Truncation and winsorization means improved the agreement in particular if the proportion truncated was high. We observed the highest performance using the Hölder mean (with parameter 0.2), followed by the geometric mean and the Lehmer mean (with parameter 0.5). If only those comparisons with expert consensus are considered (Fig 3. right panel), the agreement was generally slightly higher but the overall pattern did not change.

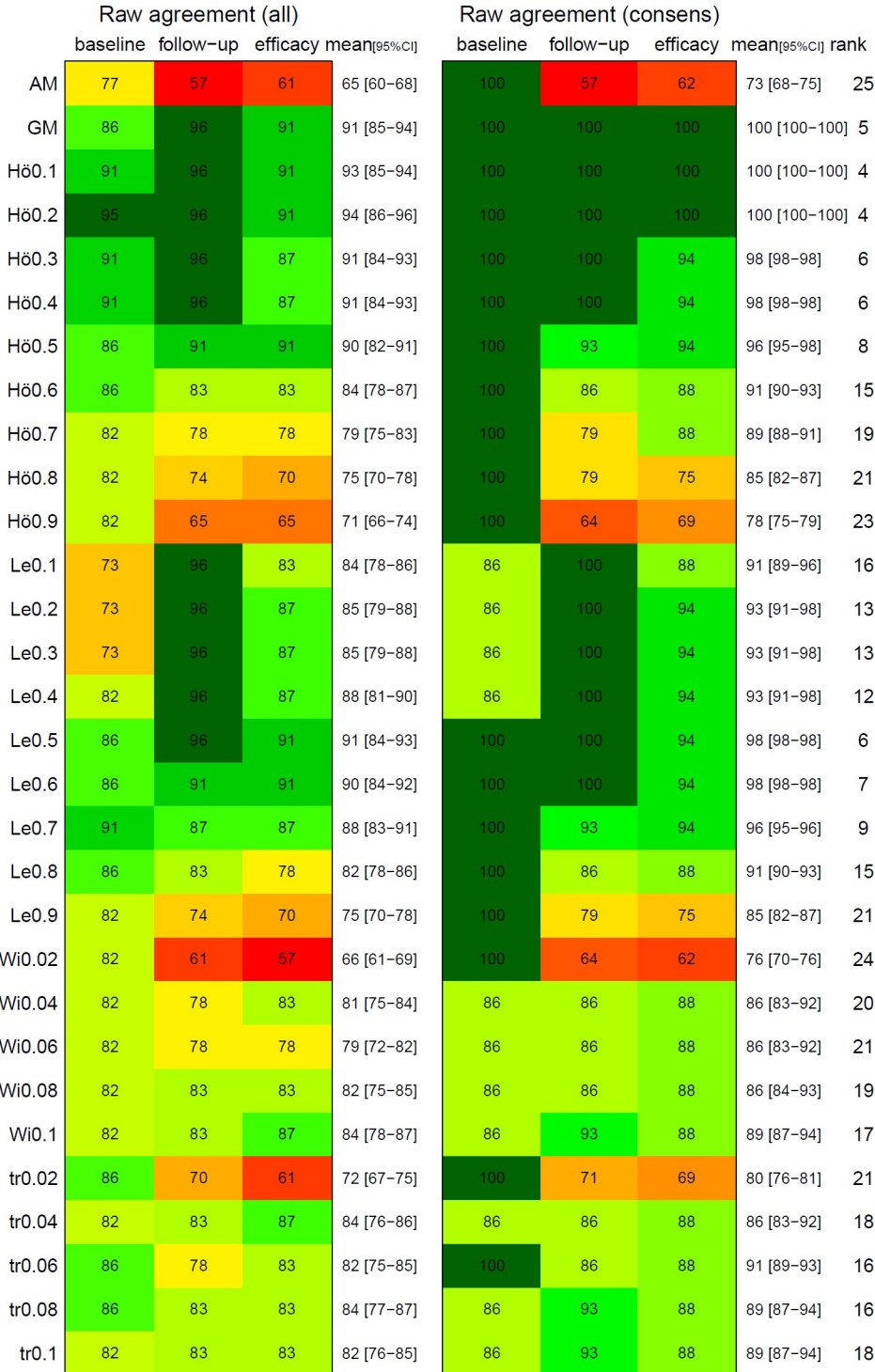

**Fig 3. Percentage agreement between experts and different means.** Raw percentage agreement between expert opinion and the calculated means for egg burden at baseline and follow-up and drug efficacy (superiority of a certain trial arm). Both, expert opinion and calculated means were dichotomized into 'A > B' and 'B > A'. Number of trial arm comparisons N: left panel: bl = 22, fu = 22, ef = 23; right panel: bl = 7, fu = 14, ef = 16. AM: arithmetic mean, GM: geometric mean, Hö: Hölder mean, Le: Lehmer mean, Wi: winsorized mean, tr: truncated mean. Numbers behind Hö/ Le indicate parameter p, numbers behind Wi/tr denote proportion discarded/replaced. The rank denotes the rounded row mean rank. All: simple majority definition, consensus: absolute majority criterion, more than 50% of experts favor one arm, i.e. only those comparisons marked with footnote a) in Fig 2 are considered. S2 File explains how Fig 2 and Fig 3 are related.

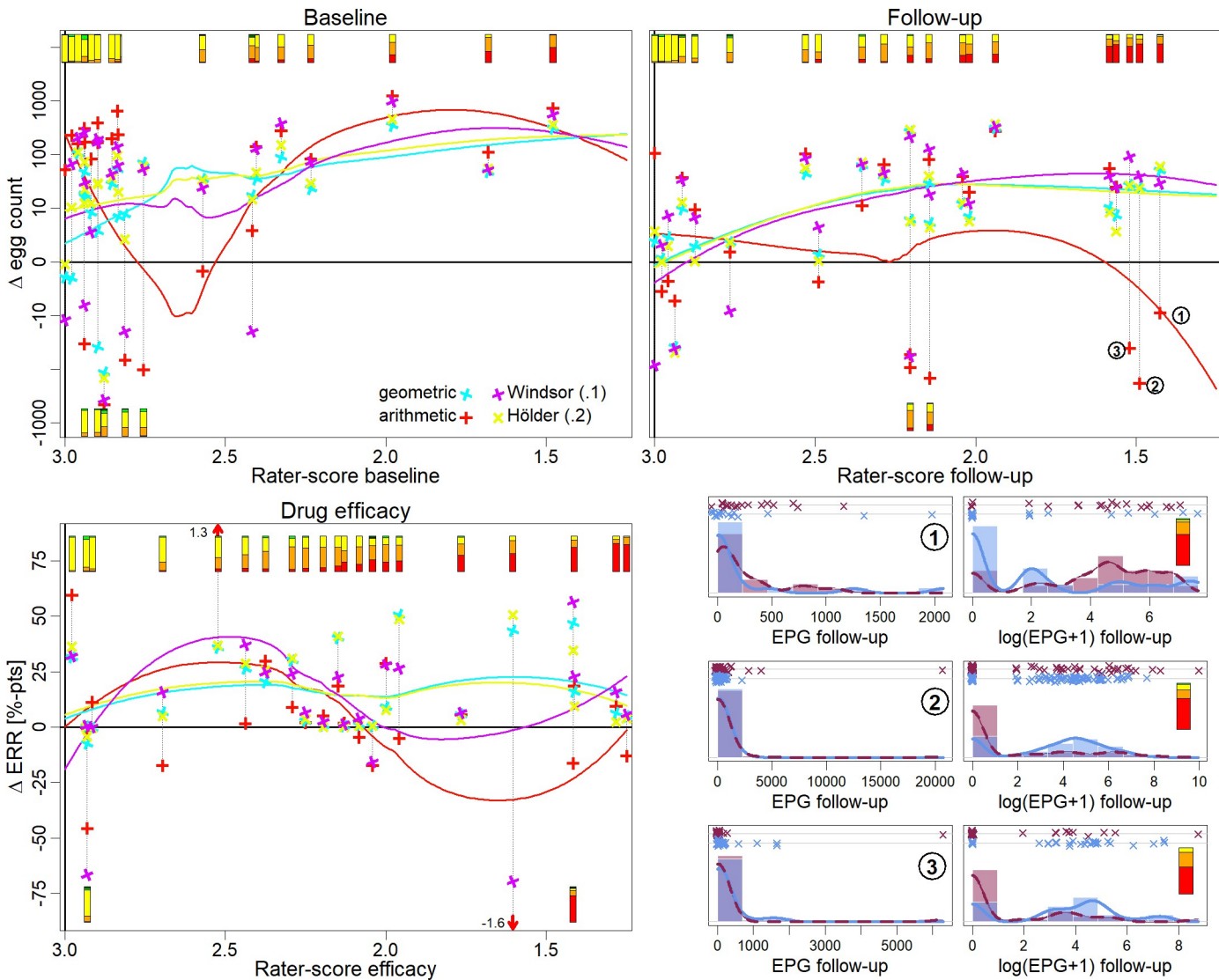

**Fig 4. Relationship between the calculated difference among 2 trial arms estimated by different means and experts' rating scores.** The symbols in the first 3 panels show the association between the rater scores and the differences in egg counts or egg reductions between trial arm pairs calculated by 4 different means (different means are represented by different colors). The lines represent the corresponding loess smoothing lines. The bar plots at the top show the experts' rating scores in the same way as in Fig 2. Some bar plots were placed at the bottom to avoid over plotting. Note, that rater scores (and bar plots) which favored arm B have been converted to favor arm A, e.g. a rating score of 4 would be converted to a score of 2 (a score of 3 indicates no difference between the trial arms). In 3 comparisons at follow up (numbered 1 to 3 in the top right panel) the estimates were especially strong diverging. The corresponding raw data are presented as strip plot and histogram in the bottom right panel. S2 File explains how Fig 2 and Fig 4 are related.

## In-depths investigation of selected means

The performance of the geometric, arithmetic, winsorized (trimmed at 10%) and Hölder (parameter 0.2) mean was explored in more detail. The arithmetic and geometric means were selected, since they are currently most commonly used and the winsorized and Hölder mean, because they showed a good performance. The relationship between experts' rating scores and the difference among the means between trial arms are presented in Fig 4.

For baseline, all means showed a correlation with the rating-scores. However, at rating scores close to 3 (indicating no difference among trial arms) all means showed considerable variability. With respect to the follow up judgments, the arithmetic mean showed the poorest

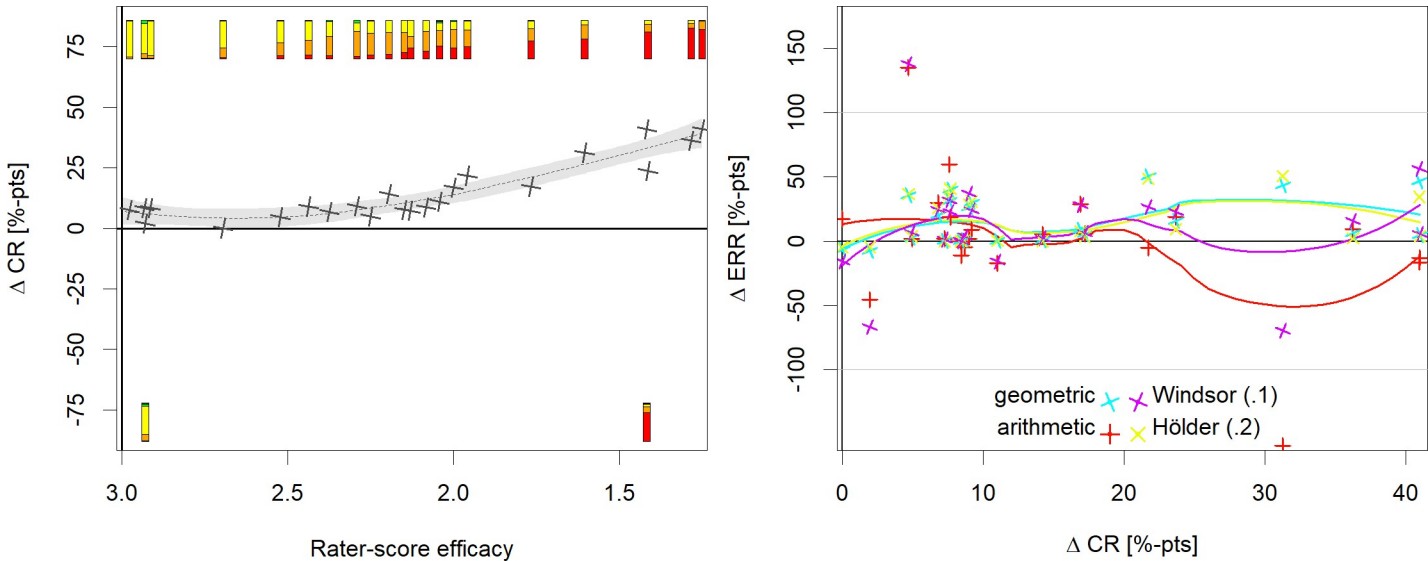

**Fig 5. Relationship between rater scores, means and cure rates.** Differences between ERRs and CRs in percentage points. Lines and shaded areas represent the loess smoothing line and the corresponding 95% confidence band. Grey crosses and the dotted line represent the experts' score and its corresponding loess smoothing line.

performance. In three of the five comparisons with rating scores below 1.75, the arithmetic mean found the opposite trial arm to be associated with a higher egg burden. In all three cases a single outlier was responsible for this result (Fig 4, lower right panel). A similar picture was observed for the drug efficacy judgments. In three of the five comparisons with rating scores below 1.75, the arithmetic mean favored the other drug. In one case the arithmetic mean estimated the difference in ERRs as 160% in opposition to the rating scores. However, this was a small trial with only 17 and 19 participants in the trial arms. Consequently, the arithmetic mean performed better in the sensitivity analysis, where small trials were excluded (S3 Fig in S3 File) but showed still the poorest performance among the four investigated means.

## Sensitivity analysis

After excluding the three trial arm pairs with less than 30 participants per arm, no noteworthy influence on the results was observed. Agreement was generally somewhat higher. One exception was the results of the Winsorized mean, which performed better in this scenario (S3 Fig in S3 File). We explored in addition the association of expert opinion and ERRs with differences in CRs. Expert opinion correlated strongly with differences in CRs whereas the correlation between winsorized and arithmetic mean ERRs and difference in CRs was again weak (Fig 5). Further sensitivity analyses using weighted lowess smoother (with weights proportional to the number of subjects in the trial arms) and with scaled differences in the ERRs (i.e. the most extreme value was considered as the minimum or maximum (S5 Fig in S3 File) supported the findings from the main analysis.

## Discussion

We calculated the egg burden and drug efficacy from several clinical drug trials against helminth infections using different types of means. The performance of the different types of means was assessed by calculating their agreement with expert opinion. From all investigated means the arithmetic mean showed the worst performance, which was sometimes not much higher than chance agreement.

The poor performance of the arithmetic mean in our study was in all scenarios related to the presence of a single outlier. Outliers might be more common in human drug trials compared to population epidemiological surveys or the veterinary sector because some participants might refuse to swallow all tablets, vomit after the treatment or do not adhere to treatment for other reasons and as randomized trials, especially dose-ranging trials, have usually relatively few participants in each arm. In addition, individual responses to treatment show remarkable variability, which might result in imbalance if the sample size is limited. Therefore, our results should not be extrapolated to studies with a different purpose, like large-scale program evaluations, resistance surveillance, environmental sanitation or the veterinary sector.

Olliaro et al. [26] pointed out: the best suited approach to assess drug efficacy depends on the purpose and for large scale program monitoring trends in responses and emergence of drug resistance are of primary interest, which can be more precisely assessed with individual level estimates. In this context, several modeling approaches have been proposed which have several advantages including estimating the full distribution of individual responses [27]. However, it might be challenging to specify rather sophisticated models a-priory in a statistical analysis plan as required in clinical trials.

Several simulation studies assessed the performance of different means with contrasting results [17–19]. Other studies relied on certain assumptions which, by design, favored one of the estimates, e.g. that the arithmetic mean based ERR represents the true efficacy [18] or that the egg counts follow a certain distribution [19,20]. To overcome the shortcomings of previous studies we used, for the first time, an approach, which does not rely on any assumptions and does not favor any particular estimate. The judgments of visualized paired comparisons might be hypothetical, because the helminth species is not specified, but provides a natural picture in terms of burden and drug efficacy. One could argue that the visualization is causing bias because of optical illusions but the consistency of our findings using complementary approaches—like associations with CRs–indicates that the results are sufficiently robust. We can only speculate about the reasons for the discrepancy between the expert opinion and the arithmetic mean. Some experts might consider extreme values as non-representative and ignore them; other experts might have the health burden in mind and prefer a large proportion of light infection even if a few heavy infections remain.

The geometric mean showed an overall robust performance in our study. The main advantage of the geometric mean is that it is simple to compute and that the mean is commonly applied for skewed data. However, there are also several disadvantages associated with this type of mean. The sample mean is biased and underestimates the population mean by a factor of $e^{var/(n^*2)}$-1 multiplied by the geometric mean. Another issue represents the fact that the geometric mean is not defined for samples that include zeros. Usually, a constant of 1 is added to each count but this constant has been criticized as being not more rational than adding any other positive number [18].

The Hölder mean slightly outperformed the geometric mean but the difference was marginal. It remains debatable if a slightly improved performance justifies the increased complexity associated with its calculation. A positive feature of the Hölder mean is that all values lie between the arithmetic and geometric mean and no modification in the presence of 0 values is required. However, in case of high CRs the estimates according to the Hölder mean could even be below the geometric mean. This is caused by the fact that–in contrast to the geometric mean–no constant is added to the zero egg counts. Considering the above stated example with 9 times 0 egg counts and one time 1000 eggs, the geometric mean would estimate a mean egg count of 1, whereas the Hölder mean (with parameter 0.2) would estimate 0.01; therefore, a higher parameter of 0.4 might be more appropriate. Likewise, the Lehmer mean requires a constant in the presence of zero values and despite it performed similar to the Hölder mean, we would not recommend its use.

In contrast to the truncated mean, the winsorized mean does not compromise the sample size and it is therefore preferable over the truncated mean. The winsorized mean with a cut-off level of 10% performed reasonably well in our study, in case the small study arms were excluded, which was highlighted in the sensitivity analysis. For obvious reasons, the estimate is not suitable for high CRs, since all CRs above 90% would result in an estimate of 100%. An additional problem might arise in case of cluster randomized trials. One needs to define if the replacement of values should be done for the entire trial arm or for each cluster separately. In this study, we applied a one sided truncation of the upper tail. It should be noted that there might be other settings where egg counts are generally quite high and zero or low egg counts represent the extreme values.

Constructing interval estimates might be challenging for several types of means in the presence of a complex study design. Confidence intervals for 2 arm superiority trials can be easily computed via bootstrapping but methods to incorporate the Hölder mean into random effect models or generalized estimating equations are currently not available. Likewise, the arithmetic mean features many statistical properties and many statistical methods rely on these properties. Further, meta-analyses on egg counts and egg reductions might become difficult to interpret if other means than the arithmetic mean are used.

There might be also biological reasons why we prefer one mean over another. The arithmetic mean of a sample is always the best estimator of the population arithmetic mean, and similarly the geometric mean of a sample is the best estimator of the population geometric mean. In environmental sanitation, we might be mainly interested in the total number of eggs shed into the environment. In this case, the arithmetic mean will be most appropriate because 'super-shedders' are of particular importance and should not be considered as outliers.

## Conclusion

In anthelminthic drug trials of moderate sample size, the ERR based on arithmetic mean—as recommended by current WHO guidelines—showed a poor agreement with expert opinion on drug efficacy. It should not be used as the primary outcome in human drug efficacy trials and should be always reported together with an estimate that is more robust to outliers. Of course, all estimates should be complemented by their corresponding confidence intervals. We recommend extending the WHO guidelines to include aspects of clinical trials besides recommendations for programme monitoring.

## Supporting information

**S1 File. Example questionnaire.**
(PDF)

**S2 File. Example explaining agreements, scores and relationship between figures.**
(PDF)

**S3 File. Trial characteristics, sensitivity analysis and agreement among different means.**
(PDF)

## Author Contributions

**Conceptualization:** Wendelin Moser, Jennifer Keiser, Jan Hattendorf.

**Formal analysis:** Wendelin Moser, Jan Hattendorf.

**Funding acquisition:** Jennifer Keiser.

**Methodology:** Wendelin Moser, Jennifer Keiser, Jan Hattendorf.

**Software:** Jan Hattendorf.

**Supervision:** Jennifer Keiser, Jan Hattendorf.

**Visualization:** Jan Hattendorf.

**Writing – original draft:** Wendelin Moser, Jan Hattendorf.

**Writing – review & editing:** Wendelin Moser, Jennifer Keiser, Benjamin Speich, Somphou Sayasone, Stefanie Knopp, Jan Hattendorf.

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
