## [Decision Letter · Decision Letter 0]

29 Oct 2019

Dear Dr. Hattendorf:

Thank you very much for submitting your manuscript "One mean to rule them all? The arithmetic mean based egg reduction rate can be misleading when estimating anthelminthic drug efficacy in clinical trials" (#PNTD-D-19-01393) for review by PLOS Neglected Tropical Diseases. Your manuscript was fully evaluated at the editorial level and by independent peer reviewers. The reviewers appreciated the attention to an important problem, but raised some substantial concerns about the manuscript as it currently stands. These issues must be addressed before we would be willing to consider a revised version of your study. We cannot, of course, promise publication at that time.

We therefore ask you to modify the manuscript according to the review recommendations before we can consider your manuscript for acceptance. Your revisions should address the specific points made by each reviewer. 

When you are ready to resubmit, please be prepared to upload the following:

(1) A letter containing a detailed list of your responses to the review comments and a description of the changes you have made in the manuscript.

(2) Two versions of the manuscript: one with either highlights or tracked changes denoting where the text has been changed (uploaded as a "Revised Article with Changes Highlighted" file); the other a clean version (uploaded as the article file).

(3) If available, a striking still image (a new image if one is available or an existing one from within your manuscript). If your manuscript is accepted for publication, this image may be featured on our website. Images should ideally be high resolution, eye-catching, single panel images; where one is available, please use 'add file' at the time of resubmission and select 'striking image' as the file type. 

Please provide a short caption, including credits, uploaded as a separate "Other" file. If your image is from someone other than yourself, please ensure that the artist has read and agreed to the terms and conditions of the Creative Commons Attribution License at http://journals.plos.org/plosntds/s/content-license (NOTE: we cannot publish copyrighted images). 

(4) If applicable, we encourage you to add a list of accession numbers/ID numbers for genes and proteins mentioned in the text (these should be listed as a paragraph at the end of the manuscript). You can supply accession numbers for any database, so long as the database is publicly accessible and stable. Examples include LocusLink and SwissProt.

(5) To enhance the reproducibility of your results, we recommend that you deposit your laboratory protocols in protocols.io, where a protocol can be assigned its own identifier (DOI) such that it can be cited independently in the future. For instructions see http://journals.plos.org/plosntds/s/submission-guidelines#loc-methods

While revising your submission, please upload your figure files to the Preflight Analysis and Conversion Engine (PACE) digital diagnostic tool, https://pacev2.apexcovantage.com/ PACE helps ensure that figures meet PLOS requirements. To use PACE, you must first register as a user. Then, login and navigate to the UPLOAD tab, where you will find detailed instructions on how to use the tool. If you encounter any issues or have any questions when using PACE, please email us at figures@plos.org.

We hope to receive your revised manuscript by Dec 28 2019 11:59PM. If you anticipate any delay in its return, we ask that you let us know the expected resubmission date by replying to this email.

To submit a revision, go to https://www.editorialmanager.com/pntd/ and log in as an Author. You will see a menu item call Submission Needing Revision. You will find your submission record there. 

Sincerely,

Matthew C Freeman, MPH, Ph.D.

Associate Editor

Sara Lustigman

Deputy Editor

Reviewer's Responses to Questions

**Key Review Criteria Required for Acceptance?**

**Methods**

-Are the objectives of the study clearly articulated with a clear testable hypothesis stated?

-Is the study design appropriate to address the stated objectives?

-Is the population clearly described and appropriate for the hypothesis being tested?

-Is the sample size sufficient to ensure adequate power to address the hypothesis being tested?

-Were correct statistical analysis used to support conclusions?

-Are there concerns about ethical or regulatory requirements being met?

Reviewer #1: (No Response)

Reviewer #2: see general comments

Reviewer #3: Please see attached reviewer comments

**Results**

-Does the analysis presented match the analysis plan?

-Are the results clearly and completely presented?

-Are the figures (Tables, Images) of sufficient quality for clarity?

Reviewer #1: (No Response)

Reviewer #2: see general comments

Reviewer #3: Please see attached reviewer comments

**Conclusions**

-Are the conclusions supported by the data presented?

-Are the limitations of analysis clearly described?

-Do the authors discuss how these data can be helpful to advance our understanding of the topic under study?

-Is public health relevance addressed?

Reviewer #1: (No Response)

Reviewer #2: see general comments

Reviewer #3: Please see attached reviewer comments

**Editorial and Data Presentation Modifications?**

Reviewer #1: (No Response)

Reviewer #2: (No Response)

Reviewer #3: Please see attached reviewer comments

**Summary and General Comments**

Reviewer #1: The authors present an interesting and (as far as I can see) novel approach towards selection between summary measures of central tendancy in calculating egg reduction rates. The rationale is clear, the methods relating to expert elicitation appear to be sound, and the paper is mostly well written. However I do have some concerns relating to the way in which the "agreement" between summary statistics and expert assessments is discussed and presented that I would like the authors to address before publication:

The paper is currently lacking a clear discussion of the fundamental difference between different types of mean at the population level, and the difference between the population mean and sample mean. When reading this paper I think the reader could get the impression that there is in reality "one true mean (to rule them all!)" but of course this is not the case; the population arithmetic mean and the population geometric mean are fundamentally different for any skewed distribution, and it is to be expected that the arithmetic mean of a sample is the best estimator of the population arithmetic mean, and similarly that the geometric mean of a sample is the best estimator of the population geometric mean. I suspect that the authors understand this well enough (from e.g. their discussion of the Dobson et al article) but I think they could do more to spell it out for the less statistically-experienced reader.

What is interesting and useful about this paper is that they make no a-priori assumption regarding which population mean is most relevant: they simply present the distributions to experts and ask which shows the best "average" (in a general sense) egg reduction. I therefore read the results more as an ellicitation exercise where the experts are tasked with selecting which of the population means best reflects their interpretation of the data; and we find that the geometric mean best reflects their qualitative feeling about the distributions. This is most likely a result of the experts tending to down-weight outliers in a similar way to the geometric mean, which is interesting. I wonder how much of this is related to the potential issue regarding refusal of treatment on lines 302-304: do the authors think that the experts were potentially dismissing extreme values as non-representative due to the potential for this type of effect? I think there is scope for a lot more discussion in this area - although it will naturally stray into social science issues with which I am not very familiar, so it could well be that some of these issues are already well described elsewhere.

However, I do think the authors are somewhat over-playing their conclusions - it is not possible based on their data to conclude that the geometric mean is in any sense a "better" measure than the arithmetic mean, rather that the experts they asked tended to down-weight outliers in a similar way to the geometric mean. This is an important point in itself - face validity is a desirable property of such a statistic - but it is not sufficient to recommend that the geometric mean be favoured in all situations. For example, the arithmetic mean remains the best indicator of the total number of eggs being excreted, and therefore the epidemiological infection pressure within the environment. I also think it is meaningless to use sample geometric means when comparing egg reduction rates to another population where arithmetic means were used: this suggests that future studies should present both arithmetic mean reduction for comparison with previous studies as well as geometric mean reduction (or some other metric) to give an estimate that may correspond more closely to an expert evaluation of the distributions. As the paper stands I think these issues could easily be misinterpreted, so I do feel that this needs to be addressed.

I also have a number of more minor comments:

- I am not sure I accept your argument that raw agreement is appropriate - what is to be lost by calculating a Kappa statistic? This is a very standard method that naturally incorporates the chance agreement probability, and also allows you another way of calculating 95% confidence intervals. Please either do this or give a better justification for not doing so.

- Line 49: I know what you mean by "problematic in skewed distribitions" but I would prefer you to describe the issue (i.e. sensitive to extreme values) directly, as it is not necessarily a "problem" in a statistical sense if it yields the best unbiased estimator of the population arithmetic mean (which it typically does)

- Line 74: I am not sure that "While" is appropriate to start this sentence - it seems like you mean something more like resistance is widespread in animal parasites and therefore is also likely in humans? Perhaps rephrase.

- Line 197: Should be "Data from six publications"

- Lne 321-322: It seems like this could use a reference

- I found appendix S1 very useful - thanks for including it. If I were to be critical, I would suggest that density plots are not a particularly useful way of representing these data as they 'smooth out' outliers - histograms (with an appropriate bin) or even empirical cumulative distribution plots may have been better. But the B+W plots are fine so I don't think it's a big problem.

Reviewer #2: Moser and colleagues assessed the agreement in interpreting drug efficacy data applying a variety of measures for central tendency. Based on the differences in agreement across the experts they conclude that the current recommended arithmetic mean should be reported together with an estimate which is more robust to outliers, e.g. geometric mean. Overall, I have three major concerns on the paper for which I would like to see an in depth response. 

First, one should not only interpret the efficacy of drugs based on a measure of central tendency only. Instead this should be interpret along the uncertainty intervals around the point estimate. I therefore strongly believe that the recommendation should not focus on including different measures of central tendency (which often give totally different point estimates), rather we need to recommend reporting the appropriate 95% confidence intervals. This is a well established practice in veterinary parasitology, but not in the human parasitology (see also WHO guidelines were conclusions on drug efficacy are based solely based on point estimates).

Second, I strongly doubt whether the applied methodology is appropriate. To me this exercise can only be assessed through a simulation study where a substantial number of datasets with varying mean and skewness of egg counts, number of outliers, sample size and true underlying drug efficacy are generated. Subsequently, the different measures of tendency are applied and there deviation from the true underlying efficacy is then assessed. This would indeed allow for more evidence-based recommendations. 

Third, the authors often make references to the field of veterinary parasitology. Although there are indeed clear similarities between the field of veterinary and human parasitology, there are some important differences which, in my opinion, are not always accurately reflected in the current draft. For example, cure rate is not used at all in veterinary parasitology to assess the efficacy of anthelmintic drugs. In contrast to WHO, who only recently recommended ERR based on the arithmetic mean, its veterinary counter part (World Association for the Advancement of Veterinary Parasitology; WAAVP), has been recommending reporting both arithmetic mean and 95% CI since 1992. I therefore would like to propose that the authors either explicitly mention these differences in to the manuscript, or re-direct the focus of the manuscript to the human field only. I would prefer the latter given the focus of the journal, the dataset used and the link to WHO guidelines. 

Minor comments

Line 67: Abbreviate soil-transmitted helminths as STHs instead of STH

Line 77: I tend to disagree that recommendation for anthelmintic drugs the primary output would be different from a clinical medicine and public health; this is in particular when drugs are not 100% and CR is affected by the analytic sensitivity of the diagnostic tool. In addition, in veterinary parasitology, CR is not recommended at all, and hence this sentence may need some revision (see also general comment). Note that reference 4 are not the WAAVP guidelines, Coles et al., 1992 Vet Parasitol are. 

Line 85: I would strongly recommend not only draw conclusions on the measure of tendency only; taken the uncertainty intervals on board would be quite essential to draw conclusions. It is therefore my belief, that this statement deviates from the fundamental approach to interpret point estimates.

Line 88: it will be important to highlight that each of these different measures of central tendency result in quite different point estimates, which will only further create confusion. Again, one should not only focus on the point estimate. 

Line 95: it is not clear to me why assumptions on true drug efficacy or egg distributions would favour one specific mean over the other one. In fact, I strongly believe that this is the only way forward (see also general comment). However, I tend to agree that the simulation studies did not include outliers, that is why it would be good to include this in newly designed simulation study. 

Line 123: I found the presentation of the egg counts (providing boxplots separately for baseline and follow-up) rather misleading. Now the experts only have some idea on the central tendency, but not on the individual variation in response across treatment arms (are high excretes at baseline still excreting most or the least eggs at follow-up). As a consequence of this are not provided with all the information to make a proper on judgement on the difference in drug efficacy across treatment arms. This might have been done on purpose, but it once more highlights the need for interpreting the point estimate with uncertainty intervals. 

Line 303: I found this sentence rather redundant, it is not really relevant here, or I mis understood the message. Do you mean that arithmetic can be used to assess infection intensity but not for drug efficacy or is a difference made between trials to designed to assess / compare drugs and trials designed to assess efficacy of drugs in large scale deworming progams?

Line 305: in general ERR would not really recommended at all to assess differences in drug efficacy in dose-response trials or in any other trial designed to assess differences. This is because the variation in point-estimates might be to large, that is why CR is probably better (given that subjects are randomized treatment arms stratifying for baseline egg counts). 

Line 310: I am not sure whether this statement holds true; reporting ERR based on group means does not exclude variation due to individual response, as long as the uncertainty intervals around the point estimate are provided. As a consequence of this, the following argumentation on the use of simple measures of central tendency over complex models does not hold true anymore (see also first major comment).

Line 348 – 353: overall I largely disagree with the drawn conclusions. The authors are providing the wrong message. In stead of providing 2 different measures of central tendency, which only creates confusion, one should emphasize the need for reporting uncertainty intervals, which may already reflect the impact of outliers.

Reviewer #3: Please see attached reviewer comments

PLOS authors have the option to publish the peer review history of their article (what does this mean?). If published, this will include your full peer review and any attached files.

Reviewer #1: Yes: Matthew Denwood

Reviewer #2: No

Reviewer #3: Yes: Luc E. Coffeng

---

## [Decision Letter · Decision Letter 1]

23 Feb 2020

Dear Dr. Hattendorf,

Thank you very much for submitting your manuscript "One mean to rule them all? The arithmetic mean based egg reduction rate can be misleading when estimating anthelminthic drug efficacy in clinical trials" for consideration at PLOS Neglected Tropical Diseases. As with all papers reviewed by the journal, your manuscript was reviewed by members of the editorial board and by several independent reviewers. The reviewers appreciated the attention to an important topic. Based on the reviews, we are likely to accept this manuscript for publication, providing that you modify the manuscript according to the review recommendations. 

Sincerely,

Matthew C Freeman, MPH, Ph.D.

Associate Editor

Sara Lustigman

Deputy Editor

Reviewer's Responses to Questions

**Key Review Criteria Required for Acceptance?**

**Methods**

-Are the objectives of the study clearly articulated with a clear testable hypothesis stated?

-Is the study design appropriate to address the stated objectives?

-Is the population clearly described and appropriate for the hypothesis being tested?

-Is the sample size sufficient to ensure adequate power to address the hypothesis being tested?

-Were correct statistical analysis used to support conclusions?

-Are there concerns about ethical or regulatory requirements being met?

Reviewer #3: See reviewer attachment

**Results**

-Does the analysis presented match the analysis plan?

-Are the results clearly and completely presented?

-Are the figures (Tables, Images) of sufficient quality for clarity?

Reviewer #3: See reviewer attachment

**Conclusions**

-Are the conclusions supported by the data presented?

-Are the limitations of analysis clearly described?

-Do the authors discuss how these data can be helpful to advance our understanding of the topic under study?

-Is public health relevance addressed?

Reviewer #3: See reviewer attachment

**Editorial and Data Presentation Modifications?**

Reviewer #3: See reviewer attachment

**Summary and General Comments**

Reviewer #1: Thanks for considering my suggestions - I think your additions have greatly improved the manuscript, and it is now much clearer from reading the main paper what exactly you mean by "performance". 

I have only a couple of further comments in relation to the abstract (which I think is currently identical to the previous version):

Lines 34-36 currently states: "Among all investigated means, the arithmetic mean showed poorest performance and agreed with the expert opinion in only 64% (bootstrap confidence interval: 60−68)."

I think this is still misleading as it suggests that it BOTH showed poorest performance AND agreed with expert opinion less than the others, whereas in fact you judge performance based on expert opinion. How about something like:

 "Among all investigated means, the arithmetic mean showed the poorest performance with only 64% agreement with expert opinion (bootstrap confidence interval: 60−68)."

Line 39: I think the comma should be between 'CI: 78-87)' and 'but' rather than between 'well' and 'after'

Line 42: The wording of "necessarily provide reliable estimates in anthelminthic efficacy" is also a bit vague and potentially misleading. How about something like:

 "necessarily rank anthelminthic efficacies in the same order as might be obtained from expert evaluation of the same data"

Line 60: Could you make it more explicit exactly what you mean by performance here, eg: "...showed the poorest performance in terms of agreement with expert opinion"

I think that the authors can be trusted to address these minor issues without need for a further review.

Reviewer #3: See reviewer attachment

PLOS authors have the option to publish the peer review history of their article (what does this mean?). If published, this will include your full peer review and any attached files.

Reviewer #1: Yes: Matthew Denwood

Reviewer #3: Yes: Luc E. Coffeng
---

## [Editor Report · Decision Letter 2]

1 Mar 2020

Dear Dr. Hattendorf,

We are pleased to inform you that your manuscript 'One mean to rule them all? The arithmetic mean based egg reduction rate can be misleading when estimating anthelminthic drug efficacy in clinical trials' has been provisionally accepted for publication in PLOS Neglected Tropical Diseases.

Before your manuscript can be formally accepted you will need to complete some formatting changes, which you will receive in a follow up email. A member of our team will be in touch within two working days with a set of requests.

Best regards,

Matthew C Freeman, MPH, Ph.D.

Associate Editor

Sara Lustigman

Deputy Editor

---

## [Editor Report · Acceptance letter]

19 Mar 2020

Dear Dr. Hattendorf,

We are delighted to inform you that your manuscript, "One mean to rule them all? The arithmetic mean based egg reduction rate can be misleading when estimating anthelminthic drug efficacy in clinical trials," has been formally accepted for publication in PLOS Neglected Tropical Diseases.

Best regards,

Serap Aksoy

Editor-in-Chief

Shaden Kamhawi

Editor-in-Chief
